# Does Adjuvant Chemotherapy Benefit Patients with T4 N0 Colon Cancer?

**DOI:** 10.3390/medicina60081372

**Published:** 2024-08-22

**Authors:** Goncagul Akdag, Deniz Isik, Akif Dogan, Sedat Yildirim, Oguzcan Kinikoglu, Alper Topal, Sila Oksuz, Ezgi Turkoglu, Heves Surmeli, Tugba Basoglu, Ozlem Nuray Sever, Hatice Odabas, Mahmut Emre Yildirim, Nedim Turan

**Affiliations:** Dr Lütfi Kırdar Kartal Eğitim ve Araştırma Hastanesi, 34865 Istanbul, Turkey; dnz.1984@yahoo.com (D.I.); akif.dogan1@saglik.gov.tr (A.D.); rezansedat@hotmail.com (S.Y.); ogokinikoglu@yahoo.com (O.K.); dralpertopal@gmail.com (A.T.); silaoksuz@gmail.com (S.O.); ezgiturk_90@hotmail.com (E.T.); hevessurmeli@hotmail.com (H.S.); basoglutugba@gmail.com (T.B.); ozlem.sever@hotmail.com (O.N.S.); odabashatice@yahoo.com (H.O.); emremahmutyildirim@gmail.com (M.E.Y.); turan.nedim@hotmail.com (N.T.)

**Keywords:** colon cancer, T4N0 stage, adjuvant treatment

## Abstract

*Background and Objectives*: Colorectal cancer (CRC) poses a major global health challenge, with high incidence rates and ongoing treatment debates. Adjuvant chemotherapy benefits for high-risk subgroups, particularly stage II disease, remain controversial. This study seeks to clarify this issue by specifically examining the impact of adjuvant chemotherapy on disease-free survival (DFS) and overall survival (OS) in patients diagnosed with T4 colon cancer. *Materials and Methods*: This retrospective study analyzed patients undergoing radical surgery for T4 colon cancer between 2002 and 2023. *Results*: Our study of 184 pT4 pN0 colon cancer patients revealed that 79.3% received adjuvant chemotherapy. Multivariate analysis demonstrated significant DFS improvement: a 60% reduction in risk for those who received adjuvant therapy (0.40 95% CI: 0.25–0.62, *p* < 0.001). Lymphovascular invasion (LVI) and adjuvant treatment were also significantly associated with OS. Adjuvant treatment reduced mortality by 60% (HR: 0.40, 95% CI: 0.23–0.68, *p* = 0.001). Patients with LVI had a 1.9-fold increase in mortality (HR: 1.94, 95% CI: 1.17–3.20, *p* = 0.011). These findings underscore the potential value of adjuvant chemotherapy and highlight the importance of treatment completion in managing T4 colon cancer. *Conclusions*: Our study identifies LVI and adjuvant chemotherapy as key prognostic factors in T4 colon cancer patients. These results support the consideration of adjuvant chemotherapy in this patient population.

## 1. Introduction

Colorectal cancer (CRC) is a common and deadly disease. Approximately 153,000 new cases are diagnosed annually in the United States alone [1]. The burden is significant globally, with Turkey seeing 21,718 new CRC diagnoses in 2022. For both Turkish men and women, CRC remains the third most prevalent cancer, with incidence rates of 27.6 and 23.2 per 100,000, respectively. At diagnosis, 39.1% of cases are localized, while a concerning 21.3% have already reached the metastatic stage [2].

While certain factors worsen prognosis, the role of chemotherapy in high-risk stage II disease remains a point of debate. Conflicting studies leave the benefits uncertain, with some suggesting no advantage [3,4,5,6,7] and others indicating potential gains for specific patient groups [4,8,9,10]. A large meta-analysis, the MOSAIC study, demonstrated that adjuvant therapy improved survival in stage II patients with localized perforation, obstruction, pT4 lesions, and those who had fewer than 12 lymph nodes dissected. In contrast, adjuvant chemotherapy did not show survival benefits in patients with lymphovascular invasion [LVI], perineural invasion [PNI], or poorly differentiated tumors [11].

The eighth edition of the American Joint Committee on Cancer (AJCC) staging manual defines the T4 stage in colorectal cancer as the tumor infiltration of either the visceral peritoneum (T4a) or adjacent organs (T4b) [12]. These T stages are associated with a poorer prognosis in colorectal cancer, with an increased risk of recurrence and lower survival rates [13,14,15]. One study highlights the huge differences in 5-year survival for node-negative colorectal cancer: 92.5% for stage I (T1/T2N0M0), 83.6% for stage IIA (T3N0M0), 76.3% for stage IIB (T4aN0M0), and only 58.8% for stage IIC (T4bN0M0) [13]. This drastic decline emphasizes the urgent need to pinpoint prognostic factors affecting disease-free survival (DFS) and OS in patients with T4 colorectal cancer. This study aims to clarify such factors within this specific T4 colon cancer population.

## 2. Materials and Methods

### 2.1. Design of This Study and Criteria for Patient Inclusion

A total of 2894 patients with colon cancer who had undergone curative resection between March 2002 and March 2023 were examined in this study, and 184 patients admitted to Kartal Dr. Lütfi Kırdar City Hospital who underwent curative surgery for primary colorectal cancer and had pT4a (visceral peritoneal involvement) or pT4b (adherence to adjacent organs or structures) adenocancer based on operative pathology results were included in this study. Data sourced from the hospital database and medical records were retrospectively analyzed. Patient data included age, sex, comorbidities, ECOG performance status, family history, tumor localization, type of surgery, histopathology, T4 stage, grade, number of removed lymph nodes, LVI, PNI, and microsatellite instability [MSI]. Additionally, adjuvant treatment regimens, duration of adjuvant treatment, recurrence status and metastasectomy were analyzed.

A special file was created for all patients in the oncology department, and consent was obtained for the use of their data in future scientific studies. Patients who did not consent were not included in this study. Access to personal information in patient files was restricted to only those physicians involved in the scientific publication of this study. All procedures conducted in this study involving human participants adhered to ethical standards set by institutional and national research committees. This research complied with the principles of the 1964 Helsinki Declaration and its subsequent amendments or equivalent ethical norms. Approval for this study was obtained from the Ethics/Institutional Review Board of Kartal Dr. Lütfi Kırdar City Hospital in Istanbul, Turkey, under approval number 2023/514/256/16 on 28 August 2023.

This study included patients who met the criteria of pathologically confirmed T4 N0 colon cancer according to the 8th edition TNM and had undergone curative resection surgery. Exclusion criteria included rectal cancer, which is a completely separate disease with different treatment and follow-up approaches; insufficient medical records, which would not have contributed to evaluating survival and relapse factors; and patients with a follow-up period of 1 year or less. Additionally, positive surgical margins, which would require postoperative re-resection or radiotherapy, and patients with distant metastasis at diagnosis, who have significantly different systemic treatments compared to localized disease, were excluded from this study. Patients with secondary cancers were also excluded, as they would have hindered transparent survival analysis due to their impact on postoperative pathology reports. Clinical T4b tumors that received neoadjuvant treatment were not included in this study because neoadjuvant therapy given before the tumor will show the pathological stage as lower than it actually is, and our study specifically aimed at the factors affecting recurrence in patients given adjuvant therapy. Thus, an attempt was made to ensure homogeneity in this study. 

### 2.2. Adjuvant Chemotherapy and Patient Follow-Ups

Patients who did not have a condition that would prevent them from taking oxaliplatin, such as poor performance status, neuropathy, allergy, or renal dysfunction, were given combined therapy as an adjuvant treatment. Patients who could not take oxaliplatin were given single-drug treatments. Patients in the chemotherapy group received the modified FOLFOX regimen (folinic acid, fluorouracil, and oxaliplatin), capecitabine alone or the XELOX regimen (capecitabine and oxaliplatin), and the FUFA regimen (fluorouracil plus leucovorin). Adjuvant treatment was tried for 6 months unless the patient refused or had to interrupt the treatment due to side effects. The follow-up period started following the completion of adjuvant treatment. Clinical examinations, chest X-rays, ultrasounds of the abdomen, and measurements of serum cancer markers were performed 3 and 6 months after surgery and repeated every 6 months throughout the first 5 years. Chest and abdominal computed tomography was carried out 1 year after surgery and repeated annually.

### 2.3. Data Collection and Outcome Assessment

The information collected encompassed various factors such as age, gender, tumor location, pathological and histological features, TNM stage, adjuvant treatment status, treatment duration, adjuvant chemotherapy regimens, and recurrence status. The categorization of cases into the ‘Right colon’ group was based on tumors located from the cecum to the hepatic flexure. In contrast, the ‘Left colon’ group included cases with tumors located from the descending colon to the splenic flexure. Tumors between the splenic and hepatic flexures were classified under the transverse colon. The primary outcome measures examined in this study were DFS and OS.

### 2.4. Statistical Analysis

Categorical variables are presented as counts (percentages), while age is reported as means ± standard deviations. For numerical variables between two independent conditions, Student’s *t*-test was applied if the data were normally distributed; otherwise, the Mann–Whitney U test was employed. The chi-square test was used to evaluate the statistical significance of any group differences.

Time-to-event analyses were conducted for DFS and OS, and Kaplan–Meier estimates were plotted. Univariate Cox regression analyses were employed to assess variables that were potential predictors for DFS and OS. Factors with *p* < 0.05 in univariate analyses were included in multivariate analyses using the Cox model. Hazard ratios (HRs) and their respective 95% confidence intervals (CIs) were calculated. A *p*-value of <0.05 was considered statistically significant. The data underwent statistical analysis using SPSS 22.0 software (SPSS Inc., Chicago, IL, USA).

## 3. Results

### 3.1. Baseline Clinic and Demographic Findings

One hundred and eighty-four patients were identified as meeting the inclusion criteria during the study period. The median age of the study group was 61.0 ± 13.3 years. One hundred and seven patients (58.2%) were men. The most common tumor localization was left colon: 118 (64.5%) patients. Eleven (6.0%) patients had a transverse colon tumor. A total of 45 (24.4%) patients underwent emergency surgical procedures. Following curative surgery, 146 (79.3%) patients received adjuvant chemotherapy. A total of 95 (51.6%) patients experienced recurrence, and metastasectomy was performed in 23 (24.2%) following recurrence. Local recurrence was observed in 14 of 95 patients who developed recurrence, while distant recurrences were seen in the remaining patients. Metastasectomy was performed in 23 patients with limited metastases in the liver and lungs. The remaining patients received systemic chemotherapy. Other patient characteristics are presented in Table 1.

### 3.2. The Logistic Regression Model for Recurrence Risk

The patients’ gender, age, type of surgery, tumor grade, histopathological subtype, presence of PNI and LVI, tumor localization, MSI status, number of removed lymph nodes, adjuvant treatment regimen, and duration of adjuvant treatment did not show a statistically significant impact on recurrence. Additionally, 67 (70.5%) patients who developed recurrence received adjuvant treatment, and this difference was statistically significant. Adequate lymph node dissection was performed in 66 (69.5%) patients with recurrence, and this difference was statistically significant (Table 2).

### 3.3. Outcomes for Disease-Free Survival

The patients’ median DFS during the follow-up period was 95.2 months. Cox regression analysis was employed to assess prognostic factors influencing DFS. When examining gender, age, type of surgery, grade, histopathological subtype, tumor localization, MSI status, and adjuvant treatment regimen, no statistically significant effect was observed. Meanwhile, the median DFS (mDFS) of patients with lymphovascular invasion was 76.1 months and without lymphovascular invasion was 135.8 months. This difference was not significant (*p* = 0.058). Despite the lack of statistical significance in the evaluation of perineural invasion status, patients without perineural invasion had a better DFS curve. The mDFS was 71.5 months for patients with insufficient lymph node dissection and 107.4 months for those with more than 12 lymph nodes removed, and no statistically significant difference was observed. While the mDFS of patients who received adjuvant therapy was 115.1 months, it was 24.2 months in patients who did not receive it (*p* < 0.001) (Figure 1).

In multivariate analysis, a 60% improvement in DFS was observed in those receiving adjuvant therapy (HR: 0.40 95% CI: 0.25–0.62, *p* = 0.000). The findings from the univariate and multivariate analyses for DFS are presented in Table 3.

### 3.4. Outcomes for Overall Survival

The patients’ median OS (mOS) during the follow-up period was 110.6 months. Cox regression analysis was used to assess the prognostic factors impacting OS. When examining gender, age, number of removed lymph nodes, type of surgery, histopathological subtype, grade, presence of PNI, tumor localization, MSI status, adjuvant treatment regimen, and duration, no statistically significant effect on OS was observed.

However, LVI and adjuvant treatment status showed statistically significant associations with OS (*p* = 0.009, *p* = 0.000, respectively). The mOS for patients with LVI was 97.5 months (95% CI: 64.6–130.3), whereas for those without LVI, it was 180.1 months (95% CI: 67.7–292.5, *p* = 0.009) (Figure 2). The mOS for patients receiving adjuvant treatment was 168.8 months (95% CI: 104.9–232.7), while for those not receiving adjuvant treatment, it was 45.7 months (95% CI: 32.0–59.3, *p* = 0.000) (Figure 3).

Two important independent variables were determined to affect OS in multivariate analysis: the presence of LVI and adjuvant treatment status. In multivariate analysis, a 60% improvement in OS was observed in those receiving adjuvant therapy (HR: 0.40, 95% CI: 0.23–0.68, *p* = 0.001), and lymphovascular invasion was found to be an independent poor prognostic factor (HR: 1.94, 95% CI: 1.17–3.20, *p* = 0.011). Table 4 summarizes the findings from the univariate and multivariate analyses for OS.

## 4. Discussion

In this study, the recurrence rate following curative resection was 51.6%, which is higher than reported in the literature [16]. Adjuvant chemotherapy was given to 79.3% of patients at the same time. However, compared to the literature’s data (41.7–72.8%) [14,16,17,18,19,20], its implementation was higher. Our clinical experience of T4 tumors being more aggressive is the reason for the literature’s comparatively higher usage of adjuvant treatment. Despite the higher percentage of the two-drug regimen usage in our T4 colon cancer patients, we observed a higher recurrence rate compared to the literature. This could be attributed to the aggressive nature of T4 tumors, differences in patient characteristics, such as overall health and tumor biology, variations in surgical techniques or postoperative care, and possibly a more rigorous follow-up methodology. These factors may collectively contribute to the increased recurrence observed in our study. Similar to our data, in a study conducted by Naxerova et al., it was stated that node-negative T4 tumors may be resistant to chemotherapy due to direct peritoneal involvement and that recurrence rates may be higher despite long-term chemotherapy [21].

Although adjuvant chemotherapy is well known for its benefits in advanced colon cancer, little research has been conducted on how adjuvant chemotherapy affects the prognosis of T4 colon cancer. Teufel and colleagues reported significant benefits for adjuvant chemotherapy regarding recurrence, recurrence-free survival, and OS in patients with T4N0M0 colon cancer in their population-based multicenter cohort analysis [9]. Makari et al. did not find a statistically significant correlation between adjuvant chemotherapy and DFS in their study of the effect of adjuvant chemotherapy on the survival of patients with T4 colon cancer. Nonetheless, adjuvant treatment showed a substantial advantage for OS in their multivariate analysis [14]. The absolute benefit of an additional three-month regimen with oxaliplatin is more significant for stage II disease than what is observed for stage III disease (3.3% in DFS), according to the preliminary report of the IDEA collaboration data. It was suggested to balance this with the increased toxicity risk associated with six months of treatment [22]. Conflicting results continue to be obtained regarding the effect of adjuvant therapy on survival. Risk factors reported to be effective in recurrence vary from study to study and are not generally accepted. Furthermore, studies showing that patients with risk factors benefit from adjuvant therapy are not convincing enough. In our study, receiving adjuvant chemotherapy had a statistical effect on DFS, and a 60% improvement in DFS was observed in those receiving adjuvant therapy (HR: 0.40 95% CI: 0.25–0.62, *p* < 0.001). While the mDFS of patients who received adjuvant therapy was 115.1 months, it was 24.2 months in patients who did not receive it (*p* < 0.001). The reason why other risk factors other than adjuvant chemotherapy were not significant can be explained by the fact that we applied long-term, intensive treatment to most of our patients, because relapse was observed in a significantly shorter time in those who did not receive adjuvant chemotherapy.

Patients with stage II disease who have certain clinicopathologic features have been linked to a poorer prognosis: T4 primary [23,24]; high-grade/poorly differentiated histology [3,25] (including signet ring, mucinous, and undifferentiated tumors); LVI [24,26,27,28]; PNI [8,24,29,30,31]; close, indeterminate, or positive margins; and inadequately sampled lymph nodes [32,33]. A study conducted at Memorial Sloan Kettering on 448 individuals who had stage II colon cancer and received curative resection without postoperative chemotherapy examined the impact of clinicopathologic variables on outcomes. Only three independent variables were shown to affect prognosis in multivariate analysis: T4 primary, preoperative CEA > 5 ng/mL, and LVI or PNI. Patients with zero, one, or two or more unfavorable prognostic markers had five-year disease-specific survival rates of 95, 85, and 57 percent, respectively [24]. In our study, when factors affecting OS were examined, LVI and adjuvant treatment status showed statistically significant associations with OS (*p* = 0.009, *p* < 0.001, respectively). In the multivariate analysis, LVI and adjuvant treatment status were effective independent variables affecting OS. Patients with mucinous adenocarcinoma histology in our study had a more prolonged OS, which was thought to be related to the small number of patients and intensive adjuvant treatment.

Adjuvant treatment reduced mortality by 60% (HR: 0.40, 95% CI: 0.23–0.68, *p* = 0.001). Patients with LVI had a 1.9-fold increase in mortality (HR: 1.94, 95% CI: 1.17–3.20, *p* = 0.011). However, PNI, age, stage, or other characteristics did not significantly affect OS.

The limitations of this study are that it was a single-center, retrospective study. In addition, the cases were not randomly selected, and the distribution between T4a and T4b was unequal. The number of patients in the T4b group was small; hence, no statistically significant independent prognostic factors could be identified in the T4b group. Furthermore, the cause of mortality during the research period was unknown, and the OS was not verified as cancer-specific survival. Other limitations of this study include a very long time period, the lack of a sufficient number of patients, and heterogeneity in surgical techniques and adjuvant treatment. The results of this study add to the body of information in this field and highlight the prognostic factors affecting DFS and OS in T4 colon cancer despite these limitations that prevent the results from being generalized.

## 5. Conclusions

In conclusion, it was determined that LVI and adjuvant chemotherapy were prognostic factors in patients with T4 colon cancer. Adjuvant chemotherapy may improve prognosis in these patients.

### Limitations

Our study has several limitations. Firstly, it is a retrospective study. Secondly, our study is a single-center experience with a relatively small number of patients, which may have affected the results of some analyses.

## Figures and Tables

**Figure 1 medicina-60-01372-f001:**
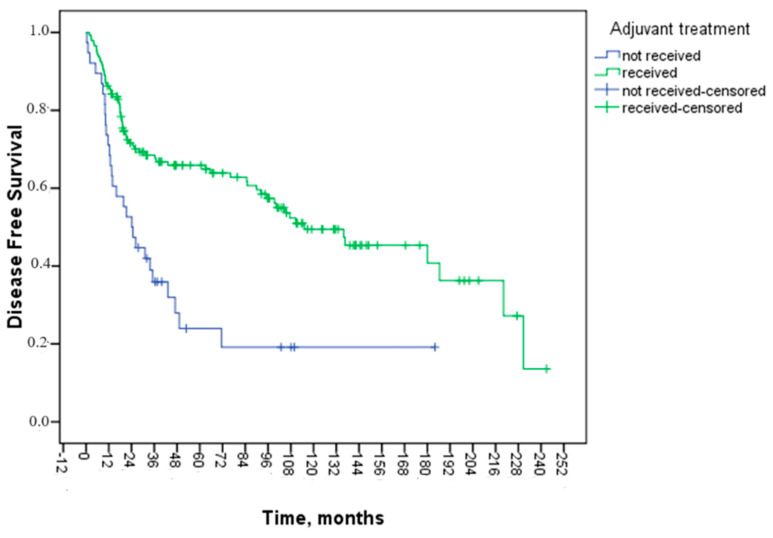
Kaplan–Meier curve of disease-free survival analysis according to adjuvant treatment status.

**Figure 2 medicina-60-01372-f002:**
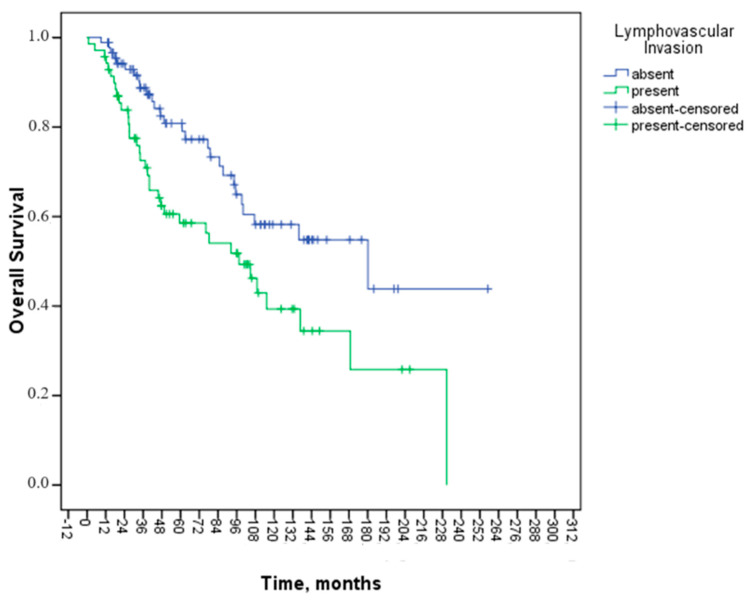
Kaplan–Meier curve of overall survival analysis according to lymphovascular invasion status.

**Figure 3 medicina-60-01372-f003:**
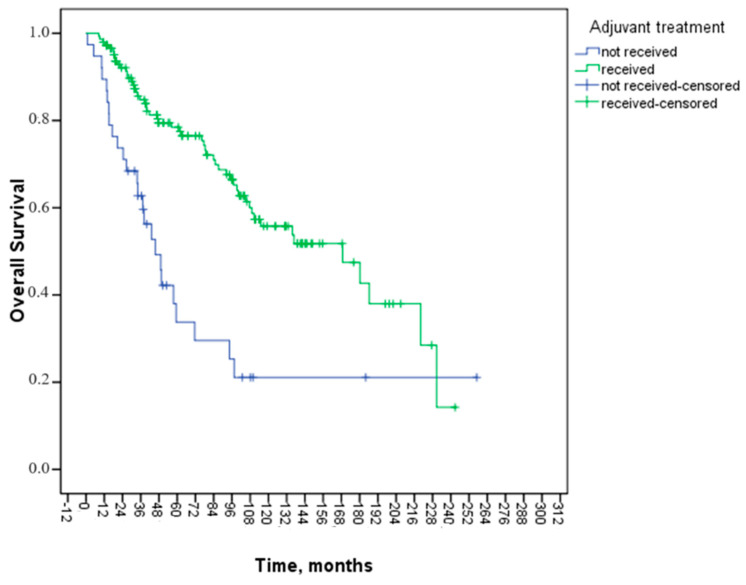
Kaplan–Meier curve of overall survival analysis according to adjuvant treatment status.

**Table 1 medicina-60-01372-t001:** Patient and tumor characteristics based on adjuvant chemotherapy (CTX) in complete cohort.

Variables (*n* = 184)	Total *n*, (%)	+CTX (146) *n*, %	−CTX (38) *n*, %
**Age (median)**	61.0 ± 13.3	60.0 ± 12.4	69.5 ± 15.7
**Gender**			
Female	77 (41.8)	63 (43.2)	14 (36.8)
Male	107 (58.2)	83 (56.8)	24 (63.2)
**Comorbidity**			
Present	84 (45.7)	64 (43.8)	20 (52.6)
Absent	100 (54.3)	82 (56.2)	18 (47.4)
**Type of surgery (*n* = 118)**			
Emergency	45 (38.1)	39 (38.2)	6 (37.5)
Elective	73 (61.9)	63 (61.8)	10 (62.5)
**Histopathology**			
Adenocarcinoma	145 (78.8)	113 (77.4)	32 (84.2)
Mucinous adenocarcinoma	37 (20.1)	32 (21.9)	5 (13.2)
Signet ring cell adenocarcinoma	2 (1.1)	1 (0.7)	1 (2.6)
**T stage**			
T4a	125 (67.9)	96 (65.8)	29 (76.3)
T4b	59 (32.1)	50 (34.2)	9 (23.7)
**Grade (*n* = 160)**			
Grade 1	23 (14.3)	17 (13.2)	6 (19.4)
Grade 2	118 (73.8)	98 (76.0)	20 (64.5)
Grade 3	19 (11.9)	14 (10.8)	5 (16.1)
**Lymphovascular invasion (*n* = 160)**			
Present	70 (43.8)	53 (41.7)	17 (51.5)
Absent	90 (56.2)	74 (58.3)	16 (48.5)
**Perineural invasion (*n* = 156)**			
Present	70 (44.9)	54 (43.2)	16 (51.6)
Absent	86 (55.1)	71 (56.8)	15 (48.4)
**Tumor budding (*n* = 30)**			
Present	9 (30.0)	7 (30.4)	2 (28.6)
Absent	21 (70.0)	16 (69.6)	5 (71.4)
**Number of removed lymph nodes**			
<12	42 (22.8)	33 (22.6)	9 (23.7)
≥12	142 (77.2)	113 (77.4)	29 (76.3)
**Microsatellite instability (*n* = 46)**			
MSI-L	28 (60.9)	23 (62.2)	5 (55.6)
MSI-S	5 (10.9)	4 (10.8)	1 (11.1)
MSI-H	13 (28.3)	10 (27.0)	3 (33.3)
**Tumor location (*n* = 183)**			
Right	54 (29.5)	43 (29.7)	11 (28.9)
Left	118 (64.5)	91 (62.8)	27 (71.1)
Transverse	11 (6.0)	11 (7.5)	-
**ECOG PS (*n* = 113)**			
0	63 (55.8)	57 (58.2)	6 (40.0)
1	34 (30.0)	32 (32.7)	2 (13.3)
2	14 (12.4)	8 (8.1)	6 (40.0)
3	2 (1.8)	1 (1.0)	1 (6.7)
**Adjuvant chemotherapy regimen (** ** *n* ** **= 146)**			
XELOX	59 (40.4)	59 (40.4)	-
FOLFOX	28 (19.2)	28 (19.2)	-
Capecitabine	17 (11.6)	17 (11.6)	-
5-FU	42 (28.8)	42 (28.8)	-
**Duration of adjuvant treatment (** ** *n* ** **= 126)**			
<3 months	18 (14.2)	18 (14.2)	-
3–6 months	23 (18.3)	23 (18.3)	-
6 months	85 (67.5)	85 (67.5)	-
**Recurrence**			
Yes	95 (51.6)	67 (45.9)	28 (73.7)
No	89 (48.4)	79 (54.1)	10 (26.3)
**Metastasectomy after recurrence**			
Yes	23 (24.2)	18	5
No	72 (75.8)	49	23
**Current status**			
Alive	106 (57.6)	93 (63.7)	13 (34.2)
Dead	78 (42.4)	53 (36.3)	25 (65.8)

MSI: microsatellite instability; T stage, Tumor stage; ECOG PS: Eastern Corporation Oncology Group Performance Status; FOLFOX: folinic acid, fluorouracil, and oxaliplatin; XELOX: capecitabine and oxaliplatin; 5-FU: 5-Fluorourasil.

**Table 2 medicina-60-01372-t002:** An analysis of clinicopathological and treatment characteristics in the recurrence group of patients.

	Recurrent *n* = 95	*p*
**Gender**		0.456
Female	37	
Male	58	
**Comorbidity**		0.659
Present	45	
Absent	50	
**Type of surgery (*n* = 35)**		0.304
Emergency	16	
Elective	19	
**Histopathology**		0.076
Adenocarcinoma	79	
Mucinous adenocarcinoma	14	
Signet ring cell adenocarcinoma	2	
**T stage**		0.876
T4a	64	
T4b	31	
**Grade (*n* = 81)**		0.782
Grade 1	11	
Grade 2	59	
Grade 3	11	
**Lymphovascular invasion (*n* = 79)**		0.111
Present	40	
Absent	39	
**Perineural invasion (*n* = 76)**		0.147
Present	39	
Absent	37	
**Tumor location**		0.285
Right	25	
Left	66	
Transverse	4	
**Number of removed lymph nodes**		0.013 *
<12	29	
≥12	66	
**Microsatellite instability (*n* = 14)**		0.877
MSI-L	8	
MSI-S	2	
MSI-H	4	
**Age**		0.236
≥60 years	57	
<60 years	38	
**ECOG PS (*n* = 34)**		0.540
0	19	
1	8	
2	6	
3	1	
**Adjuvant treatment**		0.003 *
Received	67	
Did not receive	28	
**Adjuvant chemotherapy regimen (*n* = 67)**		0.027
XELOX	20	
FOLFOX	15	
Capecitabine	6	
5-FU	26	
**Duration of adjuvant treatment (*n* = 57)**		0.849
<3 months	9	
3–6 months	11	
6 months	37	

MSI: microsatellite instability; T stage, Tumor stage; ECOG PS: Eastern Corporation Oncology Group Performance Status; FOLFOX: folinic acid, fluorouracil, and oxaliplatin; XELOX: capecitabine and oxaliplatin; 5-FU: 5-Fluorourasil. *p*-values indicating statistical significance are denoted in bold. * A logistic regression model was employed for both univariate and multivariate analyses to evaluate prognostic factors influencing recurrence.

**Table 3 medicina-60-01372-t003:** Analysis of patients for DFS according to clinicopathological factors.

Variable	Median DFS (Months) (95% CI)	Univariate Analysis *p* *	Multivariate Analysis HR (95%CI), *p*
**Gender**		0.206	
Female	180.1 (43.2–317.0)		
Male	95.2 (43.5–146.9)		
**Histopathology**		0.098	
Adenocarcinoma	84.9 (37.8–132.1)		
Mucinous adenocarcinoma	220.2 (-)		
Signet ring cell adenocarcinoma	19.3 (0.0–40.7)		
**T stage**		0.844	
T4a	95.2 (59.8–130.6)		
T4b	92.3 (0.0–194.6)		
**Tumor location**		0.604	
Right	92.3 (-)		
Left	89.8 (44.7–135.0)		
Transverse	186.4 (-)		
**Lymphovascular invasion**		0.058	
Present	76.1 (13.3–139.0)		
Absent	135.8 (83.8–187.8)		
**Perineural invasion**		0.082	
Present	49.2 (0.0–118.5)		
Absent	135.8 (89.5–182.1)		
**Number of removed lymph nodes**		0.292	
<12	71.5 (0.0–146.5)		
≥12	107.4 (72.3–142.6)		
**Adjuvant treatment**		0.000	0.40 (0.25–0.62) 0.000 *
Received	115.1 (64.2–166.0)		
Did not receive	24.2 (14.3–34.0)		
**Duration of adjuvant treatment**		0.209	
<3 months	84.9 (0.0–192.7)		
3–6 months	62.9 (-)		
6 months	136.6 (61.1–212.0)		
**Grade**		0.476	
Grade 1	71.5 (-)		
Grade 2	104.9 (64.3–145.4)		
Grade 3	89.8 (0.0–204.6)		

DFS: disease-free survival; CI: confidence interval; HR: hazard ratio. * Kaplan–Meier. Utilizing the Cox proportional hazards model, an analysis was conducted to assess the impact of prognostic factors on disease-free survival.

**Table 4 medicina-60-01372-t004:** Analysis of patients for OS according to clinicopathological factors.

Variable	Median OS (Months) (95% CI)	Univariate Analysis *p* *	Multivariate Analysis HR (95%CI), *p*
**Gender**		0.422	
Female	180.1 (76.1–284.1)		
Male	109.0 (92.6–125.4)		
**Histopathology**		0.425	
Adenocarcinoma	109.0 (74.2–143.8)		
Mucinous adenocarcinoma	220.2 (-)		
Signet ring cell adenocarcinoma	45.7 (-)		
**T stage**		0.541	
T4a	115.1 (83.6–146.6)		
T4b	99.2 (7.3–191.2)		
**Tumor location**		0.764	
Right	NR (-)		
Left	104.9 (86.7–123.0)		
Transverse	136.8 (78.8–194.8)		
**Lymphovascular invasion**		0.009 *	1.94 (1.17–3.20) 0.011 *
Present	97.5 (64.6–130.3)		
Absent	180.1 (67.7–292.5)		
**Perineural invasion**		0.055	
Present	99.2 (69.2–129.3)		
Absent	168.8 (120.4–217.2)		
**Number of removed lymph nodes**		0.224	
<12	104.9 (63.9–145.8)		
≥12	115.1 (51.3–178.9)		
**Adjuvant treatment**		0.000 *	0.40 (0.23–0.68) 0.001 *
Received	168.8 (104.9–232.7)		
Did not receive	45.7 (32.0–59.3)		
**Duration of adjuvant treatment**		0.312	
<3 months	NR (-)		
3–6 months	NR (-)		
6 months	168.8 (119.6–218.0)		
**Grade**		0.192	
Grade 1	NR (-)		
Grade 2	135.8 (86.0–185.6)		
Grade 3	97.0 (2.3–191.7)		

HR: hazard ratio, CI: confidence interval. * The impact of prognostic factors on survival was analyzed using the Cox proportional hazards model.

## Data Availability

The data presented in this study are available on request from the corresponding author.

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
