# Peer review of "Does Adjuvant Chemotherapy Benefit Patients with T4 N0 Colon Cancer?"

_medicina, 2024, doi:10.3390/medicina60081372_

Round 1

Reviewer 1 Report

Comments and Suggestions for Authors

1. It would be beneficial to provide a more detailed explanation of the specific inclusion criteria, especially regarding the criteria for pathologically confirmed T4 N0 colon cancer according to the TNM 8th edition.

2. The methodology lacks clarity on the specific data collected from the hospital database and medical records. Providing details on the variables collected and their relevance to the study would enhance the transparency and reproducibility of the research.

3. Including information on how patient confidentiality was maintained and how informed consent was obtained would strengthen the ethical integrity of the study.

4. Including a brief rationale for each exclusion criterion would clarify why certain patients were excluded from the study, ensuring transparency in the selection process.

5. The conclusion lacks specificity in summarizing the key findings and their implications. The statement is quite general and does not provide a detailed recap of the results of the study. It would be beneficial to include quantitative results or effect sizes to provide a better understanding of the impact of LVI (lymphovascular invasion) and adjuvant chemotherapy on the prognosis of patients with T4 colon cancer. Also, providing a brief justification or interpretation of why LVI and adjuvant chemotherapy are identified as prognostic factors would clarify the rationale behind the conclusion.

6. Image quality is very poor.

Comments on the Quality of English Language

7.  There are grammatical errors.

Author Response

**Respond to Reviewer 1:** 

First of all, thank you for your suggestions and feedback. I am explaining the changes we made based on your recommendations:

  1. Added more explanatory sentences regarding T4 colon tumors based on the 8th edition of TNM criteria.
  2. Added an explanation about which data we collected from hospital records and our own filing systems and how these might affect recurrence in the study's methodology.
  3. Provided clarification on patient privacy and ethical consent, explaining that data collection and consent processes were reviewed in our unit's special filing system, and only researchers involved in the study had access to these records.
  4. Added explanations for why each exclusion criterion was applied.
  5. Tried to provide more detailed explanations on the effects of lymphovascular invasion and adjuvant therapy on disease-free and overall survival and added a few references on this topic.
  6. Standard SPSS database output graphics were used in the figures, but efforts were made to improve image quality.
  7. Grammar mistakes were corrected as much as possible.

Reviewer 2 Report

Comments and Suggestions for Authors

1. In the introduction section, page 1, starting line 37 to line 43. The start of this section is too similar to one that can be found from Uptodate.com which is as follows, "A systematic review and meta-analysis of 23 cohort studies and one randomized trial (the MOSAIC trial of oxaliplatin versus fluoropyrimidines alone in patients with stage III or high-risk stage II colon cancer (see 'Should an oxaliplatin-containing regimen be used?' below)) concluded that adjuvant chemotherapy improved OS in high-risk stage II colon cancer (HR for death 0.64, 95% CI 0.51-0.80) [86]. However, in subgroup analysis, benefit was limited to those with localized intestinal perforation, obstruction, pT4 lesions, or <12 sampled lymph nodes; patients with LVI, PNI, and poorly differentiated histology did not gain a significant OS benefit from adjuvant chemotherapy."

2. It would be helpful to see the data without including MSI-H/d-MMR patients. We know that this subset of patients do not benefit (and potentially have worse outcomes) when treated with single agent 5FU or capecitabine. 

3. The overall paper contributes to growing knowledge that those with pT4 disease along with LVI have higher risk of recurrence. 

4. This might be a potential limitation. For high-risk stage II disease, the current standard practice is single-agent 5FU or capecitabine for 6 months, with an option for adding oxaliplatin knowing that DFS/OS is not much better. The majority of patients received doublet chemotherapy. The patients included in the study were from 2002 to 2023, with IDEA and MOSAIC trial data being presented 2018 and 2004, respectively. 

Author Response

**Respond to Reviewer 2:** 

First of all, thank you for your suggestions and feedback. I am explaining the changes we made based on your recommendations:

- The section similar to UpToDate on lines 37-43 was rewritten to avoid any possibility of plagiarism.

- MSI status is not routinely checked in our center. Therefore, we tried to retrospectively review the MSI status of the patients we could access, but as seen in the tables, we reached a limited number of patients. Moreover, since our study covers a long time period, many of our patients were treated before the importance of MSI status in stage 2 disease was known. Therefore, we could not perform additional evaluations related to this parameter.

- We are very grateful for your comments and support.

- Our clinical experience is that T4 disease is quite aggressive, and recurrence and metastases are frequently observed. Therefore, we tended to give our patients treatment similar to that of stage 3 patients, including combination therapy for six months, which, as you pointed out, contradicts the IDEA and MOSAIC studies. The purpose of our study was to question the validity of this approach. Despite the high rates of adjuvant therapy in our center compared to the literature, we still observed frequent recurrences, which we tried to explain with the aggressive pathology of our patients' tumors, the possibility that peritoneal involvement might cause chemoresistance, and the possibility of inadequacies in our surgical techniques.

Reviewer 3 Report

Comments and Suggestions for Authors

The study aims to explore potential disease-free and overall survival predictors in patients resected for locally advanced T4 non-metastatic colonic adenocarcinoma, with a particular interest in the potential benefit of adjuvant therapy. The paper is scientifically sound, the methods are mainly correctly used (see below), and the results sustain the conclusions. Although there is no novelty to the field with this paper, the results of the present study would potentially add value to the current literature. However, the present study has a few significant limitations: the long period, including heterogeneity for surgical approach and adjuvant treatment regimens, and the relatively low number of analyzed patients.

There are a few issues that should be addressed before consideration of acceptance:

The Introduction provides some epidemiological data about CRC in the US and Turkey. If available, please consider adding a more recent reference for Turkey's epidemiology.

Please consider avoiding terms like "illuminate."

Please consider providing in Material and Methods the total number of patients with curative intent surgery for colon cancer during the analyzed period to better reflect the center's high expertise. Furthermore, please clearly state whether the analyses included only patients with adenocarcinoma, as it appears to be the case.

Why did the authors exclude patients with neoadjuvant therapies? One might find the neoadjuvant treatment applicable, particularly for T4b tumors. Please comment.

What was the protocol for the postoperative follow-up for the patients in the present cohort to detect recurrence or death to disease or other causes?

What were the allocation criteria for adjuvant therapy and the type of regimen in the present cohort?

There is no data about the present cohort's type of approach (open, minimally invasive) and anastomosis or stoma formation. Furthermore, no data about CEA serum levels are provided.

No data about the last follow-up point is shown.

Please consistently provide the data in the manuscript as numbers (percentages).

The Results state that 51.6% of patients experienced recurrence during the follow-up (the follow-up time is not provided), while 23 patients underwent metastasectomy. Please provide the type of recurrence (local only, distant metastases, or both) and treatment modalities for recurrences.  

There is no data about postoperative complications or associated severe co-morbidities. These factors could influence the allocation of adjuvant therapy or the completion of adjuvant treatment.

Please provide the p values for each comparative analysis between adjuvant and no-adjuvant chemotherapy group patients in Table 1.

There is no meaning in the two analyses for recurrence and DFS. Please consider providing only the uni and multivariate analyses for DFS, eliminating the data from Table 2. Furthermore, there are conflicting data in the two analyses: analyses based on the presence/ absence of recurrence show the number of retrieved lymph nodes as a significant predictor of recurrence in the univariate analyses. At the same time, the same parameter is not statistically significant in the DFS analyses.

Interestingly, the present study shows no impact of adjuvant therapy duration and regimens treatment on DFS and OS. Please comment and discuss referring to other previous studies.

Please modify Figure 1-3 in a more friendly format (particularly the x-axis).

Instead of p values of 0.000, please consider providing p < 0.001.

Please cite Figure 2 in the text (instead of "blure-2").

How do the authors explain the different impacts of LVI on DFS and OS?

How do the authors explain the higher rates of recurrences in the present cohort compared with other studies despite the higher rates of adjuvant therapies?

The study's main findings are that adjuvant therapy represents the only independent predictor for DFS. In contrast, LVI and adjuvant therapy are independent predictors for OS in patients with curative surgery for T4N0M0 colon adenocarcinoma. How do the authors consider that the results of the present study could be used for clinical decision-making? There is no clear conclusion on how the results of the present study would be helpful for clinical practice.

Please clearly state the study's limitations as mentioned above (the long period, including heterogeneity for surgical approach and adjuvant treatment regimens, and the relatively low number of analyzed patients).

The Discussion part should be expanded.

There is a discrepancy between the number of references cited in the text and those provided in the references list. Please put them in accordance.

Nevertheless, it would be interesting to explore potential predictors of both DFS and OS in the two subgroups of patients (T4a vs T4b) because the curative intent surgery for these subcategories is pretty different. 

Comments on the Quality of English Language

 Minor editing of the English language required

Author Response

**Respond to Reviewer 3:** 

First of all, thank you for your suggestions and feedback. I am explaining the changes we made based on your recommendations:

- Added more current epidemiological data to the introduction section.

- Made the suggested word changes.

- Clarified that all our patients had adenocarcinoma histology and explained the total number of colon cancer surgeries performed at our center during the study period.

- Since the purpose of our study is to evaluate the effectiveness of adjuvant therapy in T4 patients, patients who received neoadjuvant therapy were not included. We also thought that the pathology of patients receiving neoadjuvant therapy would show less aggressive and less advanced disease than it actually was. To increase the homogeneity of the study population, we decided not to include patients who received neoadjuvant therapy.

- Added a detailed explanation of how we followed up on patients postoperatively and monitored for recurrence.

- Our clinical experience is that T4 disease is as aggressive as stage 3 disease, so we try to administer combination therapy for six months to all suitable patients. The main goal of our study was to question the accuracy of this approach and whether it provides significant benefits to our patients.

- Since the major guidelines we follow, such as NCCN and ASCO, do not use preoperative CEA levels in the definition of high-risk stage 2 disease, and only the ESMO guidelines define it as a minor marker, we focused on pathological markers in our study. All patients included in our study were referred from our surgical clinics with R0 status and completely resolved postoperative complications, so we focused on the effectiveness of adjuvant therapy and pathological features in recurrence or survival without addressing surgical aspects.

- Follow-up durations were written more clearly.

- Corrected the numbers and percentages in the manuscript.

- Relapse patterns and treatments applied were written in more detail.

- Since Table 1 only presents demographic data of the patients, no p-values were provided. This table simply describes patient characteristics without any comparison to indicate significance.

xxxxxxx

- Most of our patients received six months of combination therapy. Since the group of patients who received single-agent therapy and shorter treatment duration was small, we attribute the lack of a statistically significant difference to this situation.

- The x-axis in Figures 12 and 3 was modified to be more visually appealing.

- The writing related to p-values was corrected.

- A reference to Figure 2 was added, and the typographical error was corrected.

- The lack of effect of lymphovascular invasion on disease-free survival was attributed to our administration of intensive adjuvant therapy to all patients. We believe this intensive therapy prevented us from seeing a difference between the groups with and without LVI. However, LVI-positive patients developed metastases much more quickly than LVI-negative patients after adjuvant therapy, resulting in shorter survival. We would like to state that this is our comment on this topic.

- We attributed the high frequency of recurrence to the aggressiveness of our patients' tumors and the possibility that limited serosal-peritoneal involvement caused chemoresistance. An explanation was attempted within the article.

- We believe this study demonstrates that lymphovascular invasion is an important indicator of aggressive histology and that even with intensive adjuvant therapy, close monitoring for disease recurrence is necessary during follow-up. We aim to emphasize that this parameter should be considered more carefully in treatment and follow-up decisions for patients.

- The limitations of the study were written more clearly.

- The discussion section was expanded.

- Corrections were made to the references.

- Since the number of patients in the T4a and T4b groups was not balanced and our total number of patients was insufficient, subgroup analyses could not be performed. The overall aim of the study was to focus on the factors influencing adjuvant therapy and recurrence in T4 disease.

Reviewer 4 Report

Comments and Suggestions for Authors

This Paper clarifies the benefits of adjuvant chemotherapy in advanced colon cancer. There are  few publications about the effects of adjuvant chemotherapy on the prognosis of T4 colon cancer.

Moreover, Adjuvant chemotherapy benefits for high-risk subgroups, particularly stage II disease, remain controversial. This study demonstrates this issue by specifically examining the impact of adjuvant chemotherapy on disease-free survival (DFS) and overall survival (OS) in patients diagnosed with T4 colon cancer. Therefore, this article is very interesting to cover this gap in knowledge.

The article is presented in a well-structured manner and clear way. Most of the cited references are recent and relevant to the main topic. There is no excessive number of self- citations. In addition, they did their best to collect a cohort of T4 colon cancer patients and collect all the clinical, Pathological, treatment and survival information about this cohort. They were honest, in mentioning the weakness of this cohort. The hypothesis in the article is well defined and their study is considered an addition to the current knowledge.

I have few comments:

In line 18, Please write Lymphovascular invasion before (LVI) as it is the first time to use this word in the article.

In line 28, It is better to use for incidence the most recent reference, please use (Siegel et al. 2024 instead of (Siegel et al. 2023)

In line 30, it is better to use more recent available reference for incidence of colon cancer in Turkey. By searching, I have found another reference more recent than the used one. It shows that 21,718 new CRC diagnosed in 2022, Please find this reference. It is better than the used old one in Turkish language.

https://gco.iarc.who.int/media/globocan/factsheets/populations/792-turkiye-fact-sheet.pdf

In line 108, In the title of the table, replace the word (tumour) by tumor

In line 120, Please re-write the title of (table 2) in a clearer way showing the contents of the table, such as Analysis of clinicopathological and treatment characteristics in the recurrence group of patients.

In Table 2, ECOG PS (n=27), can you please re-check the value of n, if we add a9+8+6+1, it will be 34.

In line 134, 136, There is a repetition of the word (although) and it is better to use different words.

In lines 132, 138,139, There is a repetition of the word (while) and it is better to use different words.

In Figure 1, 2, 3, There is no Figure legends under the figures, describing it. You need to write for example that this is a Kaplan _Meier curve or plot for ……….

In line 154, Write mOS after median OS, because you used this abbreviation mOS in the lines after line 154

Author Response

**Respond to Reviewer 4:** 

First of all, thank you for your suggestions and feedback. I am explaining the changes we made based on your recommendations:

- Added words on lines 18108 and 154.

- Changed references on lines 28 and 30.

- Made the title of Table 2 more descriptive on line 120.

- Corrected the patient count related to ECOG in Table 2.

- Reduced word repetitions on lines 132, 134, 136, 138, and 139, and used different words.

- Added explanations of the curves in Figures 12 and 3.

Round 2

Reviewer 1 Report

Comments and Suggestions for Authors

It can be accept for publication

Reviewer 3 Report

Comments and Suggestions for Authors

No further comments.

Comments on the Quality of English Language

Minor editing of English language required.